# Human PCNA Structure, Function, and Interactions

**DOI:** 10.3390/biom10040570

**Published:** 2020-04-08

**Authors:** Amaia González-Magaña, Francisco J. Blanco

**Affiliations:** 1CIC bioGUNE, Bizkaia Science and Technology Park, bld 800, 48160 Derio, Bizkaia, Spain; agonzalezm@cicbiogune.es; 2IKERBASQUE, Basque Foundation for Science, Maria Diaz de Haro 3, 6 solairua, 48013 Bilbao, Bizkaia, Spain

**Keywords:** PCNA, structure, protein interactions, DNA sliding, molecular recognition, DNA replication, DNA repair

## Abstract

Proliferating cell nuclear antigen (PCNA) is an essential factor in DNA replication and repair. It forms a homotrimeric ring that embraces the DNA and slides along it, anchoring DNA polymerases and other DNA editing enzymes. It also interacts with regulatory proteins through a sequence motif known as PCNA Interacting Protein box (PIP-box). We here review the latest contributions to knowledge regarding the structure-function relationships in human PCNA, particularly the mechanism of sliding, and of the molecular recognition of canonical and non-canonical PIP motifs. The unique binding mode of the oncogene p15 is described in detail, and the implications of the recently discovered structure of PCNA bound to polymerase δ are discussed. The study of the post-translational modifications of PCNA and its partners may yield therapeutic opportunities in cancer treatment, in addition to illuminating the way PCNA coordinates the dynamic exchange of its many partners in DNA replication and repair.

## 1. DNA Sliding Clamps

Replicative DNA polymerases are the enzymes that replicate chromosomal DNA during the S-phase of the cell cycle. They can quickly polymerize thousands of nucleotides without detaching from the genomic template [1]. This fast and processive activity is conferred to polymerases by their association with multimeric ring shaped proteins known as DNA sliding clamps. They encircle and slide along the DNA, tethering polymerases and other factors to the DNA duplex [2].The first evidence of the sliding clamp structure was obtained from the polymerase III β subunit of the *E. coli* Pol-III complex. Biochemical assays demonstrated that the β subunit interacts tightly with nicked circular plasmid, while it dissociates when the DNA is linearized [3]. DNA Sliding clamps are loaded onto DNA by clamp loaders, evolutionarily conserved proteins of the AAA+ family of ATPases [4,5]. They form pentameric complexes that bind and open the sliding clamp, placing it onto the DNA 3′ end of the primer/template-junction, in an ATP dependent process [6].

Sliding clamps are functionally and structurally similar across all living organisms, including some viruses, and can assemble in homodimeric, homotrimeric, or heterotrimeric rings, with the protomers binding each other in a head-to-tail fashion [7]. Despite the low sequence similarity between the sliding clamps of different domains of life (Figure 1), they all adopt a three-dimensional pseudo-six fold symmetry structure consisting of an outer layer of 6 β-sheets and an inner layer of 12 α-helices facing the central channel [8,9].

The bacterial β clamp is a homodimeric ring that comprises two protomers [10], each one with three topologically similar domains. By contrast, the functional equivalents of β clamp in T4 bacteriophage (gene 45 protein, gp45) and PCNA (Proliferating Cell Nuclear Antigen) in eukaryotes and archaea assemble in trimeric rings, each protomer containing two similar domains connected by an interdomain-connecting loop (IDCL) [11,12] (Figure 2).

In the T4 bacteriophage ring, the two domains are less similar to each other, and consequently, the gp45 clamp has a triangular appearance instead of the hexagonal shape of the others.

The central pore has an internal diameter of approximately 35 Å, larger than the 24 Å of the double stranded DNA helix (dsDNA) in the canonical B-form [13]. Overall, clamps are acidic proteins with net negative charges. The outer surface possesses a negative electrostatic potential, but the α-helices facing the central cavity are rich in positively charged amino acids and generate a positive electrostatic potential that allows the DNA to pass through [14] (Figure 2, right panel). The negatively charged external surface might contribute to preventing non-specific interactions, facilitating the correct disposition of the DNA inside the ring.

For several years, considerable efforts have been made to structurally assess the association between sliding clamps and DNA, as well as to understand the molecular mechanism by which these ring-shape, multimeric proteins slide along DNA. However, the weakness and lack of sequence specificity of PCNA-DNA interactions made this task difficult. The first available high resolution crystallographic structure of a sliding clamp in complex with DNA was of the *E. coli* DNA polymerase III β subunit bound to a designed 10 bp dsDNA [15] (Figure 2). In order to mimic the primed DNA strand while being copied by the polymerase, the designed dsDNA had a four-base long 5′-overhang of ssDNA on one strand. The dsDNA portion appeared inside the ring, tilted to an angle of 22°, which could be explained by the contact of the front side of the ring with the DNA, but also by the interaction of the ssDNA with another symmetry related ring in the crystal lattice. The authors proposed that the β-clamp-ssDNA contact acts as a “placeholder”, attaching the clamp at the 3′ end of the primed site and preventing it from sliding off the DNA before the initiation of replication [15].

The crystal structure of a single-chain chimera of *S. cerevisiae* PCNA bound to primed DNA was also solved, but little information could be obtained from the DNA structure due to low occupancy and the presence of disordered regions [16]. Nevertheless, the model showed dsDNA within the central cavity of the PCNA ring, facilitating contact of the negatively charged phosphate backbone of DNA with positively charged residues on the inner side of PCNA. An ssDNA overhang, in this case, was not observed.

## 2. Human PCNA Structure

Proliferating cell nuclear antigen (PCNA) was concurrently discovered by two different groups. On one hand, Miyachi et al. [17] detected an auto-antigen in the sera of some patients with systemic lupus erythematosus, and because the protein was detected in the nuclei of dividing cells, they named it PCNA [18]. On the other hand, Bravo and Celis [19] identified a protein which was synthesized during the S-phase cell cycle and called it cyclin. Further experiments showed that both were the same protein of 29 kDa, which behaves as a homotrimer in solution. The PCNA 87 kDa ring is opened by the clamp loader Replication Factor C and is placed encircling the DNA duplex in a process where ATP is hydrolyzed [20]. Extensive studies demonstrate that PCNA is an auxiliary factor for the replication polymerases δ and ε (Pol δ and Pol ε), increasing their processivity by tethering them and sliding along the double-stranded DNA helix. Especially important is its role in the synthesis of the lagging strand, acting as a platform where Pol δ, flap endonuclease 1 (FEN1), and DNA ligase I (LIG1) bind to synthesize, process, and join Okazaki fragments [21]. PCNA recruits other factors to the replication fork, participating in DNA repair (translesion synthesis, homologous recombination, mismatch repair, and nucleotide excision repair), chromatin remodeling, and cell cycle control [21]. Not all of its partners bind simultaneously to PCNA, and switching may be triggered by different mechanisms: affinity-competition, proteolysis, or post-translational modifications [2].

The first crystal structure of human PCNA was solved in 1996, bound to the C-terminal region of the cell-cycle checkpoint protein p21^WAF1/CIP1^ (PDB entry: 1AXC) [22]. However, it was not until 2004 that two structures of the human PCNA trimer alone were elucidated (PDB entries 1VYM and 1W60), corresponding to two distinct spatial groups [23]. Both structures are essentially the same but a bucking trimer is seen in one, while the other presents a planar ring, indicating a certain degree of plasticity in the spatial arrangement of the PCNA trimer. There are two distinct faces of the ring: the front face, also known as the C-face (since the carboxy-terminal ends of the protomers are located there), contains a hydrophobic pocket next to the IDCL on each protomer, where polymerases and other proteins bind; while the back face has prominent loops emerging into the solvent, and is the target for post-translational modifications that alter the clamp’s properties [6].

The solution NMR (Nuclear Magnetic Resonance) spectrum of PCNA was assigned [24], and the backbone chemical shifts and many long range NOEs (Nuclear Overhauser Effect) were measured [25]. These data were consistent with the crystallographic results, indicating that the structure in solution is the same. Moreover, size exclusion chromatography coupled to multiangle light scattering (SEC-MALS) measurements showed that the trimeric form is predominant even at high concentrations, confirming that recombinant PCNA behaves as one single trimer in solution [25]. Experiments in intact cells and cell extracts showed that PCNA can form dimers of trimers [26], but even if PCNA may exist transiently as a loosely bound dimer of trimers inside cells, its functional role is unclear.

Thermal and chemical denaturation studies indicate that although human PCNA has the same three-dimensional structure as the *S. cerevisiae* homolog, it is less stable. Human PCNA also displays increased backbone dynamics compared with yeast PCNA, especially at the helices that line the inner surface of the ring. This highly dynamic and plastic behavior of human PCNA could be an evolutionary advantage to facilitate binding to a large number of ligands [25].

The crystal structure of human PCNA bound to a 10 bp DNA duplex shows the DNA inside the channel, tilted by an angle of 15° relative to the three-fold rotation axis of the ring [27]. The PCNA-DNA interface comprises six conserved basic residues spread along five α-helices of two protomers that establish polar contacts with five consecutive phosphates of one of the DNA strands. NMR studies confirmed that primed DNA does not bind the hydrophobic pocket of PCNA.

## 3. PCNA Sliding on DNA

In order to figure out the sliding mechanism, structural data were combined with molecular dynamics simulations [28]. The simulations presented a more pronounced tilting of the DNA, at approximately 30°, and a larger portion of DNA interacting simultaneously with two protomers of the trimer as compared with the crystal structure. Still, the basic residues within the PCNA channel established short-lived interactions with the phosphates of DNA, and switched between adjacent phosphates in a non-coordinated manner. The data suggest a mechanism for PCNA sliding along the DNA: when a sufficient number of polar contacts are established with adjacent phosphate groups of DNA, the ring rotates on the DNA, resulting in advancement by one base pair. Therefore, the conserved patch of basic residues on the inner wall enables PCNA to rotationally track the helical pitch by spiral diffusion, which is named “cogwheel diffusion” (Figure 3a). This process maintains DNA-protein contacts and keeps the clamp orientation invariant relative to the DNA [29]. These results are consistent with previous single molecule studies, which showed two modes of PCNA sliding along DNA. In one mode, the clamp tracks the helical pitch of the DNA duplex, resulting in a rotation of the protein around the DNA. In the second mode, which is less common, the protein translationally diffuses at higher rates [30].

Due to the clamp symmetry, DNA can interact with three equivalent sites of PCNA. Site switching is generated by 120° rotations about the threefold axis of the ring and requires the interruption and subsequent rearrangement of the clamp–DNA interface (Figure 3b). Thus, the occasional exchange of DNA among the three positions may explain the translational component of PCNA sliding observed in single-molecule studies [30].

PCNA is expected to be present on DNA in the absence of binding partners, at least transiently. For instance, during the replication of the lagging strand, the polymerase must depart from the clamp upon completing an Okazaki fragment, leaving PCNA behind on the completed fragment. The polymerase can then associate with another clamp loaded onto a new primed site, while the PCNA left behind can bind other partners for further Okazaki fragment processing. The major sliding mechanism, cogwheel sliding, confers the PCNA ring a tilted orientation with respect to the DNA, which is believed to be important for productive encounters with the enzymes. Whether the primary cogwheel sliding mode is maintained or not when PCNA is bound to different partners could be elucidated by single-molecule experiments.

The basic residues at the sliding interface are essential to establish a defined orientation of PCNA relative to DNA, which may be required by Pol δ holoenzyme to assemble and initiate the elongation of primed Okazaki fragments [27]. Moreover, it has been reported that the sliding surface of DNA can be modulated by specific lysine acetylations to control the DNA damage response [31], and by the binding of PCNA factor p15 [32]. Lysine acetylation removes the positive charges on the inner side of PCNA and is expected to favor the translational mode over the cogwheel mode, resulting in a faster sliding with no preferred tilting. If this tilting is required for productive encounters with Pol δ holoenzyme, the elongation of primed Okazaki fragments will be compromised. Supporting this prediction is the observation that the mutations of two conserved lysines at the yeast PCNA-DNA interface completely impaired yeast polymerase δ’s function in replicating circular DNA templates [33]. The binding of p15 tightens the grip of PCNA on the DNA and likely disfavors the translational mode. Therefore, although less studied than the outer surface, the inner part of PCNA is also evolutionarily conserved, highly regulated, and crucial for PCNA’s role in the replication fork [28,33].

## 4. PCNA Binding Proteins

A large network of proteins is responsible for replicating DNA with high fidelity and for repairing DNA damage through different pathways. PCNA is a global hub in DNA metabolism that interacts with a large number of proteins involved in a variety of DNA-related processes [34]. As PCNA is a symmetric homotrimer in solution, it has three identical hydrophobic pockets to simultaneously bind different partners and coordinate a variety of functions in space and time [35]. Most of the proteins that bind PCNA either are IDPs (intrinsically disordered proteins) or have IDRs (intrinsically disordered regions). IDPs lack defined secondary and tertiary structures under physiological conditions. Although they fulfil important biological functions across all domains of life, they are more abundant in eukaryotes [36]. In particular, most transcription factors, as well as proteins involved in signal transduction, in eukaryotic organisms are predicted to be disordered or to contain disordered regions [37]. This reveals a correlation between complex cell regulation and the greater presence of IDPs/IDRs [38]. Moreover, most of the proteins associated with cancer have been identified as IDP or IDR-containing proteins (79%), which underlines the crucial roles they play in several cellular events that are altered in cancer, such as cell proliferation, DNA repair, and apoptosis [39]. The evolutionary advantage of IDPs likely lies in their plasticity that allows them to interact with many ligands and their vast regulation through post-translational modifications [40]. Disordered proteins have large accessible surface areas, which increase their ability to interact with diverse binding partners through short linear motifs (SLiMs) [41]. Many of the proteins that interact with PCNA display a characteristic SLiM known as PIP box (PCNA interacting protein-box) or an extended version called PIP degron. The canonical PIP box motif is *QXXhXXaa*, where *h* is an aliphatic hydrophobic residue (frequently I, L, or M) and *a* is an aromatic hydrophobic one (F, W, or Y), whereas X can be any amino acid [9]. The PIP degron motif targets PCNA for degradation and also harbors a basic residue (K or R) four amino acids after the second aromatic residue, as well as a TD motif just before the aromatic residues [42]. Additionally, a novel motif was identified and found to be present in several PCNA interacting proteins, named KA box, that consists of residues K-A-(A/L/I)-(A/L/Q)-x-x-(L/V) [43,44].

Structural studies of PIP box containing proteins or derived peptides in complex with human PCNA have unveiled the molecular details of the PIP motif-PCNA interface. The crystal structure of a p21 fragment bound to human PCNA was the first structural characterization of this interface (PDB code: 1AXC) [22]. Since then, several co-crystal structures of PCNA with different ligands have been solved. Overall, all PCNA interacting proteins adopt a similar conformation, which consists of an extended N-terminal region, a 3_10_ helical turn of four residues enclosed by the hydrophobic residues of the PIP box, and a C-terminal region of variable length that sometimes adopts a β strand secondary structure and interacts with the IDCL. The conserved helix inserts into the hydrophobic pocket of PCNA, whereas the glutamine sticks into the so-called Q-pocket, establishing hydrogen bonds with the backbone of PCNA [22,42]. The assignment of the human PCNA NMR spectrum [24] allowed the analysis of the ^1^H-^15^N correlation spectra of PCNA in the presence of different ligands and the calculation of the perturbations in the NMR signals. Thereby, the NMR data provided complementary or new information regarding these interactions in solution at the residue level [45].

The available structural information of PCNA in complex with different partners reveals that three ligands are able to bind simultaneously the three identical protomers of the PCNA ring. Isothermal titration calorimetry (ITC) data with different peptides are all well fitted with a model assuming one set of equivalent sites, with no evidence of binding cooperativity. Therefore, it seems that ligands compete for binding with PCNA based on their affinities, which can be modulated by post-translational modifications [9].

The intrinsically disordered protein p21 is one of the PCNA binding partners showing the highest affinity (Table 1), probably because it needs to displace other proteins to block replication in response to DNA damage. This high affinity relies on efficient hydrophobic packing, as well as electrostatic interactions with the C-terminal region of PCNA [46]. Studies with variable length peptides of p21 suggest that basic residues at the N and C-termini encompassing the PIP box protein also contribute to an increase in the binding affinity. A recent study combining experimental and computational approaches confirmed this correlation between positive patches of residues flanking the PIP box motif and a strong affinity for PCNA [41]. Moreover, FEN1 endonuclease and p68 (also known as p66, the third subunit of Pol δ) exhibit fewer basic residues than p21, and accordingly present lower affinity (Table 1). However, the comparison of the structure of PCNA with full length FEN1 and with a 20-residue long PIP fragment reveals additional contacts involving the core domain and the C-terminal region of FEN1, which increase the affinity of FEN1 by three orders of magnitude.

## 5. The Unique Interaction of PCNA with p15

PCNA associated factor p15, PAF15, p15PAF, PCLAF, or KIAA0101 (hereafter p15) was first identified as a PCNA binding factor by co-immunoprecipitation and yeast two hybrid assays [47]. It is an intrinsically disordered protein of 111 residues that is overexpressed in several types of human cancer and correlates with poor prognosis [48]. It is primarily present in the nucleus and mitochondria of the cells [49]. The expression levels of p15 vary during the cell cycle, with a major peak in the S phase, where DNA replication occurs [50]. p15 is involved in DNA replication, as well as in DNA damage bypass and cell cycle progression, by interacting with PCNA trough a canonical PIP box sequence. The degradation of p15 is mediated by the ubiquitin ligase anaphase-promoting complex/cyclosome (APC/C) that targets it for degradation. Both PCNA and p15 co-localize in the nucleus during the S phase of the cell cycle, and they are associated with chromatin. The absence of p15 reduces DNA replication (reduced number of cells in the S-phase), suggesting that it regulates PCNA processive activity [32,51]. UV irradiation up-regulates p15 expression and p15-PCNA interaction [49]. Co-immunoprecipitation experiments from pancreatic cancer cell lines indicate that p15 appears as part of DNA-replication foci together with PCNA, pol δ, and the endonuclease FEN1 [52].

The structural characterization of p15 showed that it is monomeric in solution and exhibits typical features of disordered proteins, but NMR measurements provide evidence of transient non-random structural elements in certain regions, including in the central part of p15 encompassing the PIP box sequence [53].

The molecular recognition of PCNA/p15 has been characterized by an integrative structural approach and shows a unique mode of binding to PCNA that extends outside the canonical PIP box (Figure 4A) [54]. Similarly to in other PCNA binding partners, the canonical PIP box residues of p15 (^62^QKGIGEFF^69^) bind the groove on the front face of the PCNA protomer under the IDCL. The side chain of Q62 interacts with residues A252 and A208 of PCNA (Q pocket); residues I65–F67 adopt the characteristic 3_10_ helix; and the hydrophobic trident formed by I65, F68, and F69 inserts into the hydrophobic pocket of PCNA (Figure 4A). In contrast to p21, the C-terminus of p15 does not form an intermolecular anti-parallel β-strand with the IDCL of the PCNA protomer. But what makes the structure of p15 bound to PCNA unique is that at the β-turn at residues 62–59, the N-terminus is redirected towards the central channel of the ring. Thus, the central segment of p15, where the PIP box is located, binds tightly to the front and inner sides of the ring, while the N and C tails remain disordered, at opposite sides of the ring. Crystallography with a p15 fragment, NMR on the full length protein, molecular modelling [54], and SAXS data [32] support this unusual structure where the flexible N-terminus passes through the PCNA ring (Figure 4B). Furthermore, the N-terminal tail, at the back face of the ring, rich in positively charged residues, binds DNA with a *K_D_* in the micromolar range. This suggests that p15 may regulate the velocity of PCNA sliding along the DNA. ITC experiments revealed that three molecules of p15 are able to bind the three protomers of PCNA ring in a non-cooperative manner, with a *K_D_* of 1.1 μM [54]. However, in the cell, it is unlikely to have three identical molecules occupying the three hydrophobic pockets of the ring. PCNA ligands compete for the binding of the three protomers of the ring in the cell, and the preferential binding of some proteins over others presumably depends on their affinities, as regulated by post-translational modifications, steric hindrance, and relative protein concentrations at specific points in space and time. The IDP p21, with its high affinity for PCNA, might be able to efficiently displace most of the other competitors from binding to PCNA, which underlines its predominant role in the inhibition of DNA replication and cell cycle progression [22].

Solution NMR reveals that the presence or absence of DNA does not alter the PCNA-p15 interaction, and it is thereby proposed that p15 reduces the available sliding surface of PCNA. In particular, p15 may act as a belt that loosely attaches the DNA to the clamp during synthesis by replicative Pol δ. However, this tether may be released after replication blockage to allow the entrance of translesion synthesis polymerases (TLS), to bypass the DNA lesion.

The crystallographic structure of the ternary complex of p15-PCNA-DNA shows two molecules of p15 bound to the ring of PCNA, while DNA occupies the other PIP site. This evidence was further confirmed by molecular dynamics simulations showing that when p15 binds two subunits of the PCNA ring, the DNA is able to pass through and interact with the other [32].

Although considerable progress has been made regarding the knowledge of p15′s structure and interactions during the last decade, the molecular mechanism underlying p15′s function in DNA replication and lesion bypass remains controversial. The monoubiquitination of lysine residues 15 and 24 by UHRF1 in the N-terminal tail of p15 appears to be extremely important. Recently, an extensive structural and conformational characterization of doubly monoubiquitinated p15 showed that it remains disordered and binds PCNA in the same way as non-ubiquitinated p15, but binds DNA with a 5-fold reduced affinity [55], which supports the previously proposed hypothesis of p15 modulating the velocity with which PCNA slides along DNA [54]. Moreover, calorimetry experiments revealed that the double monoubiquitination mark of p15 is recognized by the RFTS domain of DNA methyl transferase 1, indicating a role of p15 in the regulation of DNA methylation maintenance.

## 6. Simultaneous Interactions with the PCNA Ring: The Tool-Belt Model

The PCNA ring has three identical PIP-box binding sites, one on each protomer, and could simultaneously bind three different ligands. The formation of a stable Okazaki fragment maturation complex may require that different protomers of the PCNA trimer interact with Pol δ, FEN1, and DNA ligase I; mediating DNA synthesis, flap cleavage, and ligation, respectively. This complex has been called the ‘‘PCNA tool belt’’ and has been experimentally demonstrated in the case of yeast PCNA [56]. However, mutant PCNA trimers with a single binding site are still capable of Okazaki fragment maturation [57], arguing that tool belts may not be absolutely required.

The crystal structure of human PCNA bound to FEN1 in the absence of DNA shows three endonuclease molecules attached to the ring, with different relative orientations [58]. It was proposed that these conformations represent inactive complexes, with FEN1 required to swing through a flexible hinge to acquire a DNA-cleavage competent position. This was the only structure of a full-length protein bound to PCNA until the recent cryo-EM structure of PCNA-DNA-Pol δ and FEN1 [59]. The reconstruction fitting a defined FEN1 conformer comprises ~20% of the particle dataset, while the reconstruction from the entire dataset shows no significant density at the FEN1 binding site. This suggests that FEN1 may be absent from the complex in a major set of particles. In a minor set of particles, FEN1 is in such an orientation that it may bind the downstream DNA when Pol δ encounters the previous Okazaki fragment. The structure, with two of the PIP-box sites occupied by the catalytic subunit of Pol δ and by FEN1, indicates that DNA ligase I would be sterically excluded, in agreement with a sequential switching of binding partners [57].

The FEN1 PIP-box is a canonical one (Table 2), but the PIP-box of p125 that binds PCNA in the complex (^999^VGGLLAFA^1006^) deviates significantly from the canonical sequence. Emerging evidence shows that the canonical PIP box definition is becoming too narrow as other sequences diverging from the canonical PIP box binding PCNA are being described.

## 7. Revisiting the PIP Box Definition

Several non-canonical sequences have been reported in Y-family of DNA polymerases (Pol η, Pol ι, and Pol κ), where the glutamine at position 1is not conserved, which bind PCNA with relatively high affinities (Figure 5 and Table 1) [60]. Pol ι also lacks the second aromatic residue. A recent structural study shows that the fourth regulatory subunit of Pol δ, p12, binds PCNA through a highly divergent PIP box lacking the glutamine and the first aromatic residue of the PIP consensus sequence (Figure 5) [61].

An alternative PCNA binding motif has been found: the AlkB homologue PCNA interacting motif (APIM), which comprises only five amino acids with the sequence [K/R]–[F/Y/W]–[L/I/V/A]–[L/I/V/A]–[K/R], present in several cytosolic proteins [59,60]. The crystal structure of PCNA in complex with APIM peptides showed a similar binding mode to the PIP motif, despite having a highly divergent sequence from the canonical PIP box (Figure 5).

A comparative analysis of the canonical and non-canonical PIP boxes revealed that the presence of an acidic residue at position 6 appears to stabilize the 3_10_ helix through a network of intramolecular hydrogen bonds and might be required for a high affinity interaction. By contrast, the presence of positive charged residues inside the PIP sequence seems unfavorable for PCNA binding (Figure 5 and Table 1) [61].

Furthermore, a recent piece of work points out that the PIP sequence has regions overlapping with RIR (Rev1 interacting region) and MIP (Mlh1 interacting proteins) motifs [62]. All of these findings suggest that a more general class of PIP-like motifs should be considered. PCNA might be able to bind a broader class of partners, and these ligands may be capable of recognizing more than one hub. However, the exact features that determine the selectivity and the affinity of the ligands are not well understood.

To date, almost 80 PIP-like proteins have been described as interacting with PCNA in the literature, using different techniques. However, not all of these experimental techniques are equally reliable, and some of them are known to yield false positives (such as yeast two hybrid, ELISA, or co-immunoprecipitation). Only those quantitative methods that use pure protein, such as ITC, crystallography, NMR, or Electron Microscopy provide unambiguous proof of a direct interaction [9]. The available information regarding the interaction of PCNA with protein fragments is summarized in Table 2. Only those interactions that have been characterized by one or more of the techniques mentioned above have been included.

## 8. Post-Translational Modifications of Human PCNA

Many post-translational modifications of human PCNA (PTMs) have been described, but the functional significance of many of them remain to be studied (Figure 6, Table 1) [63].

Most of the 16 lysine residues of PCNA can be ubiquitinated, SUMOylated, ISGylated, NEDDylated, acetylated, or methylated [21]. PCNA can be monoubiquitinated or polyubiquitinated (K63-linked) at the K164 residue. Different types of DNA-damaging agents induce human PCNA monoubiquitination, and polyubiquitination is induced in parallel. However, monoubiquitination is also observed in unperturbed human cells, suggesting that a certain fraction of PCNA is constitutively ubiquitinated [64]. TLS does not require PCNA monoubiquitination [65], and polyubiquitination has been linked to replication fork slowing and reversal through ZRANB3 DNA translocase activity [66]. The conserved K164 residue lies on the back face of the human PCNA ring, and the ubiquitin moiety is flexible relative to PCNA, as seen by NMR [67]. PCNA SUMOylation (predominantly with SUMO1) at K164, and at other sites, prevents replication fork collapse at double stranded breaks [68]. A second SUMOylation site was described in human PCNA, K254, although its role has not yet been determined [68]. The monoubiquitination of K117 has also been reported by a quantitative proteomics study using human cells, although its exact function is still unknown [69].The ISGylation of PCNA with the ubiquitin-like protein ISG15 occurs at K164 and K168, playing a role in the termination of error-prone TLS, to prevent excessive mutagenesis [70]. As different modifications on the same residue are mutually exclusive, dynamic switching between them regulates PCNA’s interactions with other proteins. NEDDYlation at K164 antagonizes ubiquitination and regulates the recruitment of Pol η in response to oxidative DNA damage [71]. Methylation at K248 stabilizes PCNA protein levels and enhances PCNA’s interaction with FEN1 [72]. A recent study in mammalian cells shows that PCNA is also di-methylated at K110 by the EH2Z enzyme, which appears to stabilize the trimeric form of PCNA [73].

The acetylation of PCNA at several lysine residues (K13, 14, 77, and 80) by CREB-binding protein (CBP), and less efficiently by p300, fosters the removal of chromatin-bound PCNA and its proteasomal degradation during nucleotide excision repair synthesis [31].Notably, these lysines are located in the α-helices that line the inner side of the ring. Single point mutations of these lysines reduce PCNA’s stimulatory effect on DNA Pol δ. Since they are still able to interact with the polymerase, impaired DNA synthesis might be due to the slowing down of the sliding caused by the misalignment of residues contacting the DNA. Furthermore, it is known that the acetylation of K14 on the inner surface of the ring negatively regulates PCNA’s interaction with MTH2 in response to UV damage, which makes PCNA susceptible to proteosomal degradation and dissociation from DNA damage sites [21,74].

Several phosphorylation sites have been identified on PCNA [21]. Y211-phosphorylation increases the half-life of chromatin-bound PCNA and facilitates cell proliferation [75]. Y211-phosphorylated PCNA facilitates error-prone DNA replication and the suppression of the MMR (mismatch repair) mechanism, which enhances cancer development and progression [76,77]. PCNA phosphorylation in the conserved Y114 in mammals has been reported to be important in the clonal proliferation leading to adipogenesis in mice [78].

PCNA is ADP-ribosylated in response to oxidative stress, on aspartate and glutamate residues at the IDCL, potentially perturbing the binding of PIP-box containing proteins [79].

## 9. Perspectives

Over 40 years of research have illuminated the role of human PCNA on essential biological processes, and its structure has been examined in the presence of numerous partners. Still, the way it coordinates the binding of specific ligands for each function remains to be elucidated. The competition via affinity is modulated by steric effects (large binders vs. small or disordered ones), the relative protein concentrations inside the nuclei at different times and in different tissues and the possible interactions between different PCNA binding partners (which will increase their overall affinity). The importance of post-translational modifications of both PCNA and its interactors is only recently starting to be uncovered. The availability of tools to produce PCNA and its partners in modified forms for structural studies will facilitate the study of the interplay between the different combinations of post-translational modifications.

PCNA is an attractive target for cancer therapy. PIP-box binding site targeting, through small molecule inhibitors, was shown to inhibit the binding of p21 and Pol δ to PCNA and to interfere with DNA replication [80]. Peptide mimetics based on the PIP-box sequence have also been designed as PCNA inhibitors [81]. However, targeting such a promiscuous binding site may cause undesired side effects. Targeting post-translational modifications on PCNA or its interactors may instead be more specific. In fact, the inhibition of PCNA phosphorylation on Y211 was shown to inhibit the proliferation of prostate cancer cells [82]. It is thus necessary to conduct more studies on the role of the different sites of post-translational modification on PCNA and its interactors to understand how PCNA coordinates the dynamic exchange of its many partners in DNA replication and repair.

## Figures and Tables

**Figure 1 biomolecules-10-00570-f001:**
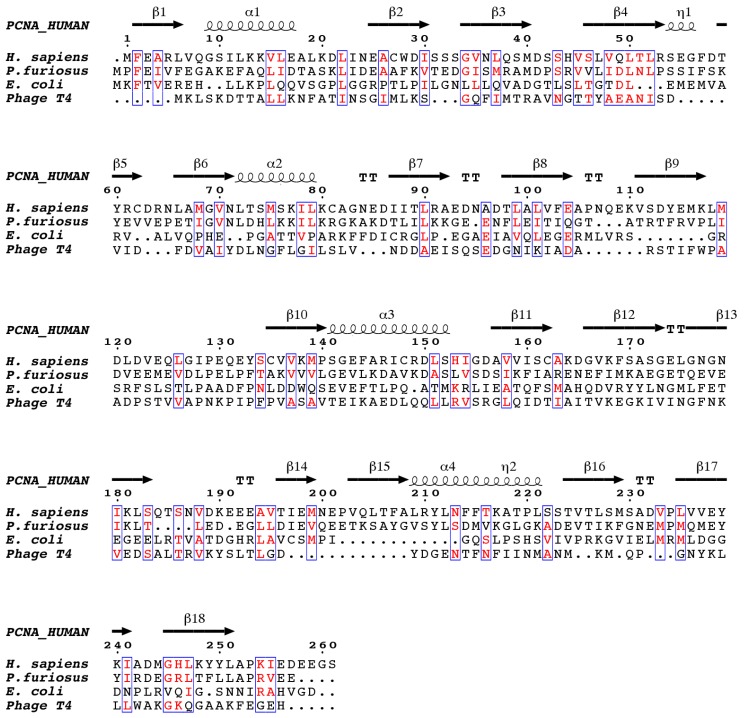
The structure-based sequence alignment of DNA sliding clamps from *H. sapiens*, *P. furiosus, E. coli*, and the gp45 gene of bacteriophage T4. The alignment was performed with chain A from each of the PDB files, which corresponds to one of the three protomers (in the case of *E. coli*, which consists of two protomers, only the N-terminal two thirds of the sequence are shown). Similar residues are colored red. Secondary structure elements corresponding to human proliferating cell nuclear antigen (PCNA) are shown above the alignment, β-strands are indicated as arrows, α-helices as spirals, and β-turns as TT. The figure was generated with ESPript.

**Figure 2 biomolecules-10-00570-f002:**
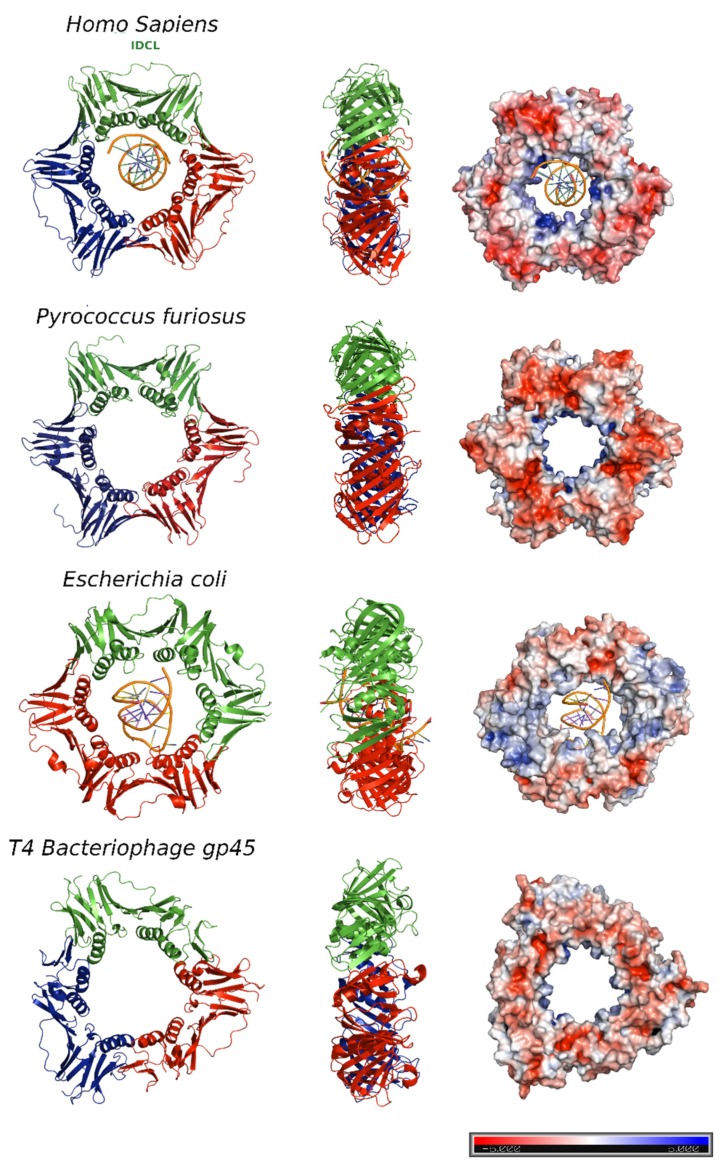
DNA sliding clamps from different organisms. The crystal structures of *Homo sapiens* PCNA (Protein Data Bank [PDB] code: 6GIS) bound to DNA, *Pyrococcus furiosus* PCNA (PDB code:1GE8), the *Escherichia coli* β clamp (PDB code: 3BEP) bound to DNA, and the gene 45 antigen of *Bacteriophage T4* (PDB code: 1CZD). For each organism, the front (left) and side views (middle) of the three-dimensional crystal structure are shown. Each protomer is represented by a different color (blue, red and green). The surface electrostatic potential is represented, with positive potential depicted in blue and negative potential in red (right). The potential varies from −5K_B_T/e to +5K_B_T/e.

**Figure 3 biomolecules-10-00570-f003:**
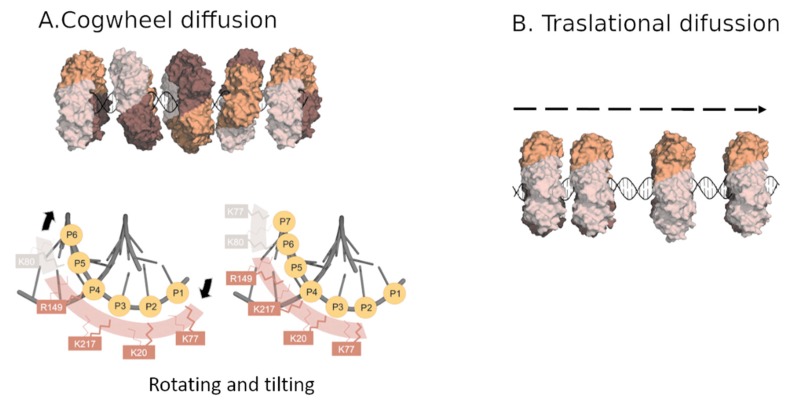
Human PCNA sliding along DNA. (**A**) Cogwheel diffusion mode: PCNA rotation tracks the DNA through a spiral motion, establishing transient interactions with DNA, which keeps the orientation of the clamp invariant relative to the DNA. The threefold rotation axis of PCNA around the DNA helical axis. In the lower panel, the evolution of PCNA–DNA contacts during cogwheel sliding is illustrated. Interacting side chains can rapidly and randomly switch between adjacent phosphates (indicated by the thin and thick lines), enabling PCNA to advance along the phosphate backbone by rotating and tilting motions. (**B**) Translational diffusion mode: PCNA travels along the DNA, without contacting the DNA. (Figure adapted from [28]).

**Figure 4 biomolecules-10-00570-f004:**
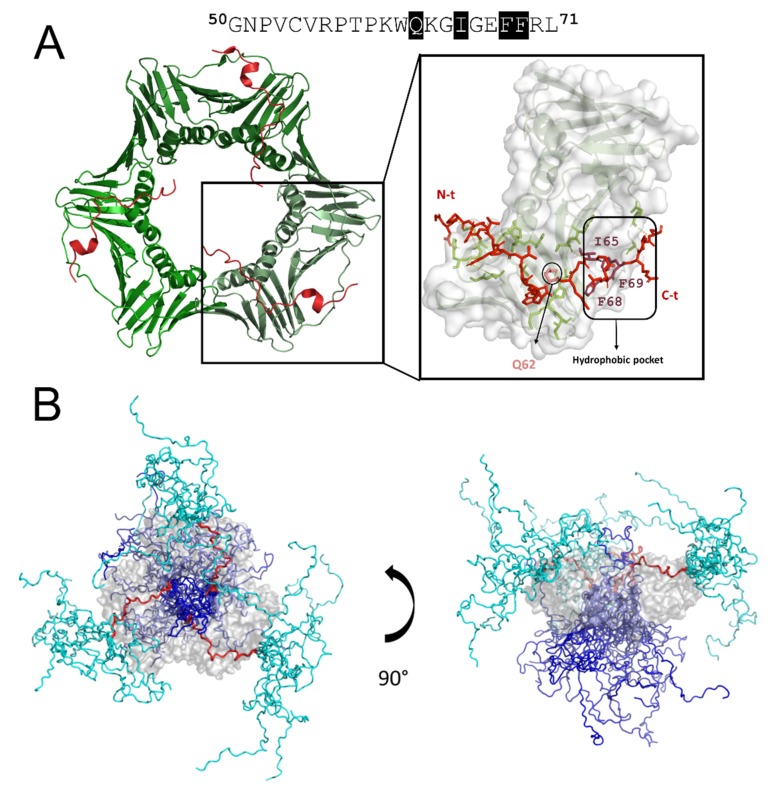
The central region of p15 binds PCNA, while the N- and C-terminal tails remain disordered. (**A**) The crystal structure of a PCNA trimer bound to three p15^41–72^ fragments (PDB code: 6GWS) in cartoon representation. PCNA is colored in a different tone of green, while p15 fragments are shown in red. On the right, a zoom view of the binding interface is shown, where polar contacts between PCNA and p15 are marked by discontinuous yellow lines. p15 peptide and PCNA are represented by sticks and cartoons, respectively. Only those residues of PCNA that interact with p15 are drawn as sticks. The surface representation of the monomer of PCNA is shown with transparency. (**B**) A model of the full-length p15–PCNA complex. Front and side views of 10 modelled structures of the p15–PCNA complex based on experimental data. PCNA is shown as a gray surface and p15 as ribbons, with the central region colored in red, and the added disordered N- and C-terminal extensions colored in cyan and blue, respectively. In one of the 10 selected complex models, one p15 N-terminus folds back towards the front face of PCNA instead of passing through the hole.

**Figure 5 biomolecules-10-00570-f005:**
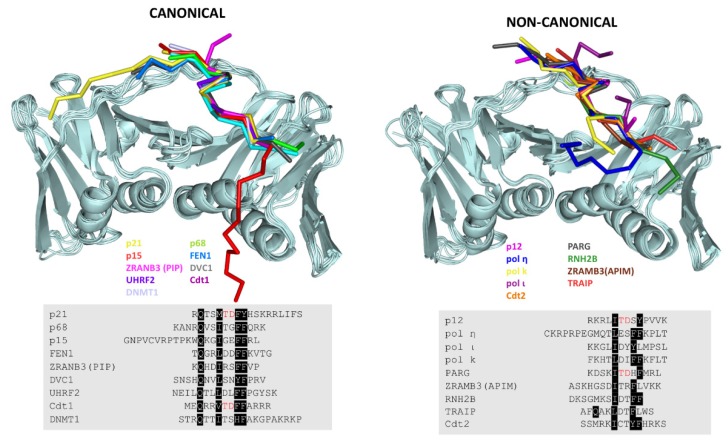
Upper panel: Superposition of the structures of canonical (**left**) and non-canonical (**right**) PCNA-interacting motifs bound to PCNA. The PCNA protomers are represented by ribbons, and the peptides are represented by their Cα traces with different colors. Lower panel: Sequence alignment of PCNA-interacting protein fragments in crystal structures bound to PCNA. Consensus residues are highlighted in black, and the TD motif of the PCNA Interacting Protein box (PIP)-degron is depicted in red. The residues shown in the alignment are those observed in the crystal structure and do not include the terminal disordered residues present in the peptides.

**Figure 6 biomolecules-10-00570-f006:**
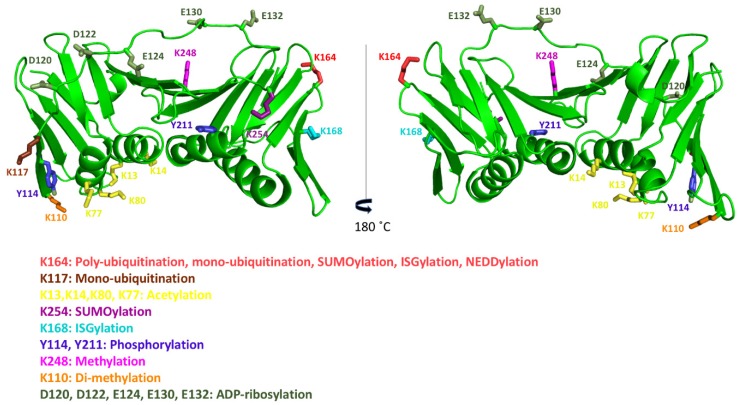
Residues that suffer post-translational modifications (PTMs) mapped on the crystal structure of human PCNA (PDB code: 1VYM). The front (left) and back (right) faces of the PCNA protomer are shown in cartoon representation. The side chains of residues undergoing different post-translational modifications are depicted by sticks in different colors.

**Table 1 biomolecules-10-00570-t001:** Post-translational modifications of PCNA.

Modification	Target Residue	Enzyme	Function	Ref.
**Monoubiquitination**	K164	Rad18, RNF8 CRL4Cdt2	Promotes DNA synthesis at damaged sites	[65]
	K117	Unknown	Unknown	[69]
**Polyubiquitination**	K164	Rad5 HLTF/SHPRH	Promotes TS	[66]
**Acetylation**	K13,K14, K77, K80	CPB/p300	Promotes PCNA degradation after NER	[31]
**SUMOylation**	K164	UBC9(E2)	Inhibits DSBs formation	[68]
	K254	Unknown	Unknown	[72]
**ISGylation**	K164, K168	EFP	Terminates TLS	[70]
**NEDDylation**	K164 *	Rad18	Regulates Pol η recruitment in DDR pathway	[71]
**Phosphorylation**	Y114 *	Unknown	Promotes adipogenesis in response to fatty diet	[76]
	Y211	EGR	Protects PCNA from degradation and inhibits MMR	[75] [77]
**Methylation**	K248	SETD8	Promotes maturation of Okazaki fragments	[72]
**Di-methylation**	K110 *	EZH2	Promotes DNA synthesis by Pol δ	[73]

* These PTMs were studied in murine cells; however, the targeted residues are conserved. TLS: Translesion synthesis polymerases; NER: Nucleotide Excision Repair; DDR: DNA Damage Response; HR: Homologous Recombination; MMR: Mismatch repair; TS: error-free template switching; DSBs: DNA double-strand breaks.

**Table 2 biomolecules-10-00570-t002:** A summary of protein-PCNA interactions validated by structural and/or quantitative methods. The conserved residues of the PIP box motif are marked in red.

Protein	Activity	Sequence	T(°C)	Kd (µM)	Method *	PDB Code	Ref
**CANONICAL**							
**p21**	**CDK1 inhibitor**	**^139^GRKRRQTSMTDFYHSKRRLIFS^160^**	30	0.080	**ITC**, XR, NMR	**1AXC**	[22]
**p68**	**Polymerase** **δ subunit**	**^451^GKANRQVSITGFFQRK^466^**	30	16	**ITC**, XR, NMR	**1U76**	[46]
**FEN1**	**Endonuclease**	**^331^SRQGSTQGRLDDFFKVTG^350^**	30	59.9	**ITC**, NMR	**1U7B**	[46]
**p15^PAF^**	**Replication/repair**	**^50^** **GNPVCVRPTPKWQKGIGEFFRLSPKDSE^77^**	25	5.56	**ITC**, XR, NMR	**4D2G** **6GWS**	[54] [32]
**ZRANB3**	**Helicase/Endonuclease**	**^511^FTHFEKEKQHDIRSFFVPQPKK^532^**	25	4.8	**ITC**,XR	**5MLO**	[83]
**DVC1**	**Adaptor protein**	**^321^SNSHQNVLSNYFPRV^336^**	25	15.55	**ITC**, XR	**5IY4**	[84]
**DNMT1**	**Methyltransferase**	**^161^STRQTTITSHFAKGPAKRKP^180^**	25	1	**ITC**, XR	**6K3A**	[85]
**UHRF2**	**E3 ubiquitin ligase**	**^784^NEILQTLLDLFFPGYSK^800^**	20	25.7	**ITC**, XR	**5ICO**	[86]
**Cdt1**	**Replication factor**	**^1^MEQRRVTDFFARRR^14^**	ND	7.2	**FP**, XR	**6QCG**	[87]
**RecQ5**	**Helicase**	**^952^KTSPGRSVKEEAQNLIRHFFHGRARCESE^980^**	35	210	**NMR**	**-**	[61]
**NON-CANONICAL**							
**pol ı**	**TLS polymerase**	**^419^CAKKGLIDYYLMPSLST^435^**	25	0.39	**SPR**,XR	**2ZVM**	[60]
**pol** **ƞ**	**TLS polymerase**	**^694^CKRPRREGMQTLESFFKPLTH^713^**	25	0.4	**SPR**,XR	**2ZVK**	[60]
**Pol ĸ**	**TLS polymerase**	**^856^CIKPNNPKHTLDIFFK^870^**	25	ND	**SPR**,XR	**2ZVL**	[60]
**ZRANB3**	**Helicase/endonuclease**	**^1058^QVRRQSLASKHGSDITRFLVKK^1079^**	25	9.24	**ITC**, XR	**5MLW** **5YD8**	[83] [88]
**PARG**	**Glycosylase**	**^402^QHGKKDSKITDHFMRLPKA^420^**	25	3.3	**ITC**, XR	**5MAV**	[89]
**p12**	**Polymerase** **δ subunit**	**^1^MGRKRLITDSYPVVKRREG^19^**	25	38	**ITC**,XR, NMR	**6HVO**	[61]
**Cdt2**	**E3 ubiquitin ligase**	**^704^SSMRKICTYFHRKS^717^**	ND	0.057	**FP**, XR	**6QC0**	[90]
**TRAIP**	**E3 ubiquitin ligase**	**^447^KQRVRVKTVPSLFQAKLDTFLWS^469^**	25	30.7	**ITC**, XR	**4ZTD**	[87]
**RNH2B**	**RNase**	**^290^DKSGMKSIDTFFGVKNKKKIGKV^312^**	-	-	XR	**3P87**	[91]

* Experimental biophysical methods that validate interactions: ITC, Isothermal Titration Calorimetry; XR, X-ray Crystallography; NMR, Nuclear Magnetic Resonance; FP, Fluorescence Polarization; SPR, Surface Plasmon Resonance. The technique used to measure the affinity is shown in bold.

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
