# Peer review of "Human PCNA Structure, Function and Interactions"

_biomolecules, 2020, doi:10.3390/biom10040570_

Round 1

Reviewer 1 Report

The review by González-Magaña and Blanco represents a wonderful effort in discussing historical, structural and functional aspects of PCNA. They also described unique binding mode between p15 and PCNA. This article would accessible not only to students, but also senior scientists after minor revisions. Comments are provided below.

I sometimes found unnecessarily space (double space?) between the words in the text. For example; L51 “and  archaea”, L66 “acids  and”, L215 “damage.  This”, L293 “PCNA  are”, L358 “PCNA.  The”, L359 “colours. Lower”, L39 “recently  proposed”, and so on. Please check carefully.

L112, The word “p1rotomer” should be “protomer”.

L139, The sentence “PCNA sliding on the PCNA” should be “PCNA sliding on the DNA”.

L293, I couldn’t found “discontinuous black lines” in Figure 4.

I couldn’t found indicated sentence “(Figure 5)” in the main text.

PCNA binding motifs was well studied in the report “A novel PCNA-binding motif identified by the panning of a random peptide display library.” (2001, Biochemistry) by Xu H et al. This paper sited “Proliferating cell nuclear antigen (PCNA): a dancer with many partners“ (Maga and Hübscher, 2003, Journal of Cell Science) and “Proliferating cell nuclear antigen is required for loading of the SMCX/KMD5C histone demethylase onto chromatin.“ (Liang et al, 2011, Epigenetics Chromatin). The author should site these articles, and describe another PCNA binding motif “KA-box”.

Reviewer 2 Report

In this review, the authors describe PCNA’s structure-function relationships, focusing on primarily sliding of PCNA on DNA and how PCNA molecularly recognizes – or is recognized by binding partners.  The review also provides a limited structural comparison across diverse organisms.  In the manuscript’s current form, the review is most helpful for, and of most interest to, an audience of a crystallographer in this field and of someone wanting to get up to date on the latest in structural interaction data that have been observed for PCNA. But, this review has the potential to reach a much broader audience if revised to make the material more accessible. The authors should also better highlight how these structural observations provide insight into biological function.  Highlighting the “big unknowns” in the field and why these unknowns are important to figure out would also strengthen the review. Clarifying what is known vs. not known would also be helpful.

Areas in which clarification would be helpful:

  1. There was a fair amount of discussion of “Cogwheel” sliding versus translational diffusion on DNA from in vitro data. What predictions could be made from either model about how PCNA functions in a cell? How could this be tested? Would PCNA ever be expected to be present on DNA and slide in the absence of other binding partners? Would binding of a partner prevent one sliding mode, but promote another?
  2. The authors comment that acetylation modulate the sliding surface of PCNA on DNA.To put this into biological context,  how? Promote or inhibit? If the acetylation event was incorporated into either sliding model, how would it be predicted to affect sliding?  What enzyme acetylates PCNA? How does loss or constitutive acetylation of this residue affect function inside the cell (what is the biological result of acetylation or loss of acetylation)?
  3. In line 107-108. …two structures of the human PCNA trimer alone were elucidated…corresponding to two distinct spatial groups (monoclinic C2 and triagonal P3, respectively).(What is that, for the non-expert?)
  4. Line 112 typo in “p1rotomer”
  5. Define terms throughout text, e.g. line 116 “NOEs”
  6. What evidence is there for different kinds of binding partners to bind different subunits of the PCNA trimer simultaneously? If these binding partners are also known to interact with each other, how does this affect the “competition via different affinities” for binding to common binding sites on PCNA?
  7. If binding partners for PCNA simply competed via differences in affinity for PCNA, what enables low affinity partners to ever bind? What else must be driving binding of low affinity interactors? E.g. Is there genetic data available on suppressors of mutants with unusually low or high affinity interactions with PCNA? How can biological information
  8. Line 201-203. A figure would be helpful.
  9. Line 227, reference on Y2H assay is missing.
  10. Line 364 and on. How are the larger post translational modifications expected to affect sliding of PCNA and what sites have been observed to be modified either alone or in combination (one same or different subunits within the PCNA trimer) in vivo? In what context do these modifications occur?  Upon specific types of DNA damage? At specific phases of the cell cycle? A figure summarizing modification sites and how they might impact the ability of partners to bind PCNA would be helpful.
  11. Clarify in Table 1 the species in which the interactions have been observed.

Reviewer 3 Report

None

Author Response

We thank the reviewer for the positive evaluation of our work.